# Automated Operational Modal Analysis for Rotating Machinery Based on Clustering Techniques

**DOI:** 10.3390/s23031665

**Published:** 2023-02-02

**Authors:** Nathali Rolon Dreher, Gustavo Chaves Storti, Tiago Henrique Machado

**Affiliations:** School of Mechanical Engineering, University of Campinas, Campinas 13083-970, Brazil

**Keywords:** automated operational modal analysis, rotating machinery, hydrodynamic bearings, rolling bearings, hierarchical clustering

## Abstract

Many parameters can be used to express a machine’s condition and to track its evolution through time, such as modal parameters extracted from vibration signals. Operational Modal Analysis (OMA), commonly used to extract modal parameters from systems under operating conditions, was successfully employed in many monitoring systems, but its application in rotating machinery is still in development due to the distinct characteristics of this system. To implement efficient monitoring systems based on OMA, it is essential to automatically extract the modal parameters, which several studies have proposed in the literature. However, these algorithms are usually developed to deal with structures that have different characteristics when compared to rotating machinery, and, therefore, work poorly or do not work with this kind of system. Thus, this paper proposes, and has as its main novelty in, a new automated algorithm to carry out modal parameter identification on rotating machinery through OMA. The proposed technique was applied in two different datasets to enable the evaluation of the robustness to different systems and test conditions. It is revealed that the proposed algorithm is suitable for the accurate extraction of frequencies and damping ratios from the stabilization diagram, for both the rotor and the foundation, and only one user defined parameter is required.

## 1. Introduction

Structural Health Monitoring (SHM) is the process of implementing a damage identification strategy for aerospace, civil, and mechanical engineering infrastructures [1]. SHM strategies have been employed in recent decades in order to improve the infrastructure’s lifetime and safety. According to Lynch, Farrar, and Michaels [2], SHM can be divided into damage detection, prognostic, and risk assessment. The first step usually consists of collecting the structure’s response over extended periods of time, followed by a data normalization for signal processing purposes, extracting damage-sensitive features, and finally, implementing a robust method for damage detection using the extracted features.

The structure’s modal parameters can be used as damage-sensitive features in damage detection since they are based on parameters that are modified in the presence of damage. The modal parameters can be extracted by modal testing, using either Experimental Modal Analysis (EMA) or Operational Modal Analysis (OMA). EMA extracts the modal parameters considering that both inputs and outputs are measured whereas OMA obtains these parameters only from the measured outputs of the system. Whereas EMA requires equipment to excite the system and needs to take the system out of operation, OMA’s premise is that the environmental loads acting upon the system excite it with an approximate white noise signal and do not require the system to go out of operation. Since the idea of SHM involves the constant monitoring of the structure, EMA is more adequate to an initial study of the modal parameters and OMA becomes an alternative to the monitoring while in operation.

There are numerous techniques to apply OMA, and one of the most employed is the Stochastic Subspace Identification (SSI). This technique has some advantages when compared to the others, such as the presence of noise truncating mechanisms based on Singular Value Decomposition (SVD), the solution of the identification problem by means of linear algebra tools, which avoids non-linear optimization problems and results in a lower computational cost, and the possibility of using weighting matrices to improve the method’s performance in the presence of noise or weakly excited modes [3]. Yet, the quality of the estimation relies on a correct choice of SSI parameters, e.g., the number of block rows, the weighting matrices, the system order, etc. Theoretically, the system order could be estimated inspecting the number of singular values that are different from zero, and the number of block rows could be determined through a direct relation between the system order and the number of outputs. However, this approach is not suitable for real data because noise is usually present in the measurements and because of the structures’ complexity. Hence, different approaches were proposed to handle real data. In order to solve the number of block rows problem, Reynders and De Roeck [4] proposed a relationship between the sampling frequency and the lowest frequency of interest, and the so-called stabilization diagram was proposed to deal with the fact that the system order is unknown and to better visualize and interpretate the technique’s results.

To build this diagram, the SSI method is performed with increasing system orders and the obtained poles are plotted on a diagram of frequency vs. system order. In other words, it is possible to identify modes that stabilize in frequency, damping ratio and mode shape with increasing orders, which usually represent physical modes of the system. Because of that, the stabilization diagram is inevitably composed of groups of the physical modes that can be identified by the group of machine learning techniques called clustering. Given that the system order can be overestimated, several spurious and mathematical poles also appear on the diagram because of the model’s attempt to better fit the experimental data, making it harder to identify the system’s modal parameters. Moreover, when the system is excited by periodic signals, as rotating machinery in operation, several poles appear on the diagram at the fundamental and harmonic frequencies of the signal, and the alignment of these poles can be misinterpreted as modes of the system. Considering this, a procedure that combines clustering techniques with other signal processing techniques to automatically interpretates the stabilization diagram is essential and can promote a better and more reliable parameter extraction and would allow the SHM to be carried out without much user interaction in the selection of the system’s modes. 

Magalhães, Cunha, and Caetano [5] proposed an algorithm for the automatic analysis of stabilization diagrams and implemented it on a set of response measurements of a bridge. First, the authors used the SSI-COV method to build the stabilization diagrams of each signal, classifying as stable all poles whose modal parameters respect the limits of variation from one order to another. Considering the block row problem preciously discussed, the authors pointed out that a separate investigation was performed to deal with it. Then, hierarchical clustering was employed to group stable poles from the stabilization diagram. The clustering was performed with the single linkage algorithm and with a similarity measure that includes both the frequency difference and the MAC (Modal Assurance Criterion) value between a pair of modes. The threshold for this distance was manually determined through the analysis of the results. Finally, an outlier analysis based on a statistical technique was performed in order to remove the extreme values of damping within each cluster. 

Reynders, Houbrechts, and De Roeck [6], on the other hand, proposed three automated steps to group the poles and applied them to response measurements collected from two different bridges. SSI-COV was used to create the stabilization diagram, and parameters such as the number of block rows and the maximum order of the stabilization diagram were manually selected. In the first step, the poles of the diagram were divided into two groups with the K-means clustering algorithm: certainly spurious modes and possibly physical modes. The K-means algorithm input was a feature vector containing as many relevant single-mode validation criteria as possible, such as frequency, damping ratio, mode shape distance measures, modal phase collinearity, mean phase deviation, etc. All poles from the certainly spurious cluster were removed from the stabilization diagram and the remaining poles were evaluated using Hard Validation Criteria (HVC), i.e., all poles whose damping ratio is out of the permissible range and that do not have a complex conjugated pair are also removed from the stabilization diagram. In the second step, the hierarchical clustering was used to group the possible physical modes, using the average linkage algorithm. Similar to Magalhães, Cunha, and Caetano [5], a similarity measure based on the frequency difference and on the MAC value between a pair of poles was considered. The similarity measure threshold was determined with an automated procedure that considers the results obtained in the previous step. In the last step, the K-means algorithm was once again employed to separate clusters of physical poles from clusters of spurious poles, being the algorithm input the number of poles in each cluster, and the clusters with highest number of elements were chosen as the physical ones. Since outliers could still be present in the clusters, the authors choose to select the pole with median damping ratio value in the cluster to represent that cluster.

Neu et al. [7] pointed out that the algorithm proposed by Reynders, Houbrechts, and De Roeck [6] is limited to approximately real vibration modes and therefore limits the damping ratio range, a premise that is not always suitable to more complex valuated mode shape systems. To overcome this issue, the authors developed a new automatic algorithm that works without any user-provided thresholds and does not place any limitations on the damping ratio or the complexity of the system under analysis. As well as the other studies mentioned, the first step of this approach was to perform the SSI-DATA for numerous system orders. Then, mathematical poles were removed through a HVC based on the real and imaginary parts of each pole. The next step was to separate the poles in two groups: the probably physical poles and the certainly mathematical ones. In this attempt, the authors used a K-means clustering technique and proposed a consistent feature vector, applying transformation and normalization techniques to highly biased vectors. Then, the hierarchical clustering technique was employed in the probably physical group with the average linkage algorithm and using as similarity measure the relationship between the frequency difference and the MAC value between a pair of poles. The similarity measure threshold was determined from the probability distribution function of probable physical modes. In addition, repeated poles of the same order inside each cluster were located and all but one were removed based on their proximity to the cluster’s centroid. The obtained clusters were separated into physical and mathematical clusters based on the number of poles in each one. Finally, the authors applied the modified Thompson Tau technique to remove outliers and the average natural frequency, damping ratio, and mode shape of each cluster were selected to represent it. The authors applied the proposed technique in measurements from a wind tunnel investigation with a composite cantilever, promoting the assessment of the algorithm’s performance with a highly damped structure and low signal-to-noise ratio conditions.

Cardoso, Cury, and Barbosa [8] proposed an algorithm inspired by the ones presented by Magalhães, Cunha, and Caetano [5] and Reynders, Houbrechts, and De Roeck [6] and applied it to data from a numerical experiment, from a laboratory experiment of a simply supported beam and from dynamic tests of a bridge. According to the authors, the main contributions of the proposed methodology rely on an innovative similarity measure that leads to a symmetric dissimilarity matrix, additional modifications regarding filtering the spurious modes with damping and Mode Phase Collinearity (MPC) criteria, and a novel cluster regrouping technique. 

More recently, automated identification of modal parameters that uses clustering techniques was studied by Fan, Li, and Hao [9], Wu et al. [10], and Mugnaini, Fragonara, and Civera [11], extending the application of the proposed algorithms on a steel frame structure, bridges, and a helicopter blade. The subject was also approached in studies presented in the International Operational Modal Analysis Conference (IOMAC) of 2022 by Amer et al. [12], Priou et al. [13], and Dreher, Storti, and Machado [14].

All abovementioned papers employed machine learning techniques to the automated identification of modal parameters and demonstrated its relevance to current research. However, most presented research focused on the development and validation of automatic algorithms for civil structures, which exhibit different characteristics when compared to rotating machinery. To the authors’ knowledge, no specific automatic algorithm for interpreting rotating machinery stabilization diagram has yet been studied. 

Regarding the application of OMA in rotating machinery, this has been and still is a subject of great importance. Since rotating machines are exposed to periodic excitation, present nonlinear behavior, closely spaced modes, among other conditions that make the application of OMA a challenge, and several authors recently investigated the applicability of OMA in rotating machines. Brandt [15] developed two methods for harmonic removal, the Frequency Domain Editing (FDE) and the Order Domain Deletion (ODD) methods. Gres et al. [16,17,18] proposed and applied a method for harmonic removal based on orthogonal projection, applying it to experimental data from a plate and a ship in operation. Gioia et al. [19] and Peeters et al. [20], on the other hand, investigated a harmonic removal technique based on cepstrum analysis, applying it to the drivetrain of a wind turbine. More recently in IOMAC of 2022, Dreher, Storti, and Machado [21] proposed a method to identify both forward and backward modes of a rotor that appeared as closely spaced modes difficult to differentiate via traditional OMA by the use of directional coordinates. In the same conference, Zivanovic et al. [22] presented a novel approach to harmonic disturbance removal in single-channel wind turbine acceleration data by means of time-variant signal modeling.

These studies emphasize the importance given to the expansion of OMA’s techniques to rotating machinery. Therefore, the objective of this work is to develop a new algorithm, based on the algorithms previously described that considers the different characteristics of rotating machinery, such as the presence of harmonics, outliers, the gyroscopic effect, and the complexity of the mode shapes, but still retains user friendliness. The main novelty of this work is the development of an algorithm that can identify the modal parameters related to the rotor, not the structure, which was not presented so far in the literature. The proposed algorithm is applied to two different datasets: response measurements of a test rig with a rotor supported by hydrodynamic bearings, and response measurements of a test rig with a rotor supported by rolling bearings, all under different operating and excitation conditions. The bearings were under healthy conditions for the generation of both datasets. Since rotating machines are also usually subjected to unideal excitation conditions with regard to OMA’s premise of white noise excitation, this study evaluates whether the automatic OMA algorithm is adequate for the identification of the rotor’s modal parameters under different excitation conditions, such as colored noise, tapping, lower sampling frequency, among others that will be further exposed, and which is another novelty of this work. 

Section 1 presented the motivation for the development of this work, together with the literature review. An overview of the approach proposed for this work is presented in Section 2, along with a brief explanation of the SSI-DATA algorithm, the explanation of the algorithm proposed for automatic modal identification, and the description of both datasets used in this work. Section 3 presents the results obtained with the proposed approach and comparisons with methodologies previously proposed in the literature are pointed out. Finally, Section 4 presents the conclusions.

## 2. Materials and Methods

### 2.1. Overwiew of the Proposed Approach

The present work was organized according to the diagram presented in Figure 1, in which the dotted areas indicate a sequence of steps that was performed repeated times in order to generate the indicated results. First, the datasets are generated. Since the same test rig is used to generate both datasets, the test setup is carried out in order to place the corresponding bearing (rolling or hydrodynamic) in the test rig. The operating condition is also defined, the rotor starts its operation, and the vibration signals corresponding to that setup and operating condition are collected. With the vibration signals of all setups and operating conditions, the acquisition of both datasets is completed. More information about the datasets is provided in Section 2.4. 

With vibration signals of both datasets, EMA and OMA analyses are performed to identify the modal parameters of the system, that is the natural frequencies, damping ratios, and mode shapes. Both the EMA and OMA methods were applied to the vibration signals of the system in order to perform the identification. The EMA analysis is performed with the Stepped Sine method to determine reference values for the rotor’s modal parameters. This is done for both the rotor supported by rolling and hydrodynamic bearings. An EMA analysis is also performed to determine the modal parameters of the rotor’s foundation. More information about the EMA analysis is also provided in Section 2.4. The OMA analysis is performed as described in Section 2.3, using the automatic OMA algorithm proposed by this work. A brief summary of this section is the application of the SSI method in a set of vibration signals to generate a stabilization diagram. From each diagram, a series of stages (including machine learning techniques) extract the modal parameters from the system that originated the set of signals. The OMA analysis is performed for all setups and operating conditions. Finally, the modal parameters extracted from the automatic OMA are compared with the reference values, and the discussions are presented in Section 3.

### 2.2. Data Driven Stochastic Subspace Identification (SSI-DATA)

Although most of the papers presented in the previous section used the Covariance Driven SSI (SSI-COV) algorithm, there are indications that the Data Driven SSI (SSI-DATA) algorithm is more precise and robust [23,24]; thus, it was chosen in this research.

The Stochastic Subspace Identification is based on the stochastic model, defined by Equation (1):(1){xk+1=Axk+𝓌kyk=Cxk+𝓋k,
in which yk∈ℝl denotes the outputs in the instant k, xk∈ℝn denotes the states in the instant k, and 𝓌k and 𝓋k denote the white gaussian noises, with zero mean, related to the process and the measurement noises, respectively. The white gaussian noises have the following covariance matrix:(2)E[(𝓌p𝓋p)(𝓌qT𝓋qT)]=(QSSTR)δpq.

The system’s order is n. Hence, the matrices dimensions are A∈ℝn×n, C∈ℝl×n, Q∈ℝn×n, S∈ℝn×l, and R∈ℝl×l.

It is assumed that the pair {A,C} is observable, which implies that all modes of the system can be observed in the outputs $y_{k}$ and, therefore, can be identified. It is also assumed that the pair $\left\{A,Q^{1/2}\right\}$ is controllable, which implies that all dynamic modes of the system are excited by the process noise.

The purpose of SSI is to use the outputs of the system to determine the systems matrices A and C, and, with them, extract the modal parameters of the system.

In order to do that, the first step is to build the output block Hankel matrix ($Y_{0|2i-1}$), that can be divided into the block Hankel matrices of the past outputs ($Y_{p}$) and the future outputs ($Y_{f}$) and is given by:
(3)$$Y_{0|2i-1}≜\left[\frac{\begin{array}{cccc} y_{0} & y_{1} & \cdots & y_{j-1}\\ \cdots & \cdots & \cdots & \cdots \\ y_{i-2} & y_{i-1} & \cdots & y_{i+j-3}\\ y_{i-1} & y_{i} & \cdots & y_{i+j-2} \end{array}}{\begin{array}{cccc} y_{i} & y_{i+1} & \cdots & y_{i+j-1}\\ y_{i+1} & y_{i+2} & \cdots & y_{i+j}\\ \cdots & \cdots & \cdots & \cdots \\ y_{2i-1} & y_{2i} & \cdots & y_{2i+j-2} \end{array}}\right]=\left(\frac{Y_{0|i-1}}{Y_{i|2i-1}}\right)=\left(\frac{Y_{p}}{Y_{f}}\right),$$
in which i is the number of block rows and j is the number of block columns. Then, the projection matrix ($O_{i}$) can be determined by the projection of the future outputs onto the past outputs and can be obtained through the QR decomposition of the output block Hankel matrix:(4)$$O_{i}=Y_{f}/Y_{p}$$

The SVD decomposition is then applied to a product of the projection matrix and weighting matrices that are selected based on the desired algorithm (Principal Component Analysis—PCA, Unweighted Principal Components—UPC, or Canonical Variate Algorithm—CVA):(5)$$W_{1}O_{i}W_{2}=USV^{T}=\left(U_{1}U_{2}\right)\left(\begin{array}{cc} S_{1} & 0\\ 0 & 0 \end{array}\right)\left(\begin{array}{c} V_{1}^{T}\\ V_{2}^{T} \end{array}\right)=U_{1}S_{1}V_{1}^{T}.$$

The projection matrix can also be expressed as the product of the extended observability matrix ($\Gamma_{i}$) and the forward Kalman filter state sequence (${\hat{X}}_{i}$):(6)$$O_{i}=\Gamma_{i}{\hat{X}}_{i}.$$

Therefore, the extended observability matrix and the state sequence are determined by:(7)$$\Gamma_{i}=W_{1}^{-1}U_{1}S_{1},$$
(8)$${\hat{X}}_{i}=\Gamma_{i}^{†}O_{i}.$$

Similar operations can be used to determine the shifted state sequence (${\hat{X}}_{i+1}$). Then, the system’s matrices A and C can be determined applying the least square method to the following equation, derived from the stochastic model (Equation (1)):(9)$$\left[\begin{array}{c} {\hat{X}}_{i+1}\\ Y_{i|i} \end{array}\right]=\left[\begin{array}{c} A\\ C \end{array}\right]{\hat{X}}_{i}+\left(\begin{array}{c} \rho_{𝓌}\\ \rho_{𝓋} \end{array}\right),$$
in which $\rho_{𝓌}$ and $\rho_{𝓋}$ are the Kalman filter residues. The modal parameters extraction, along with more details about the hole procedure, can be found in [23].

### 2.3. Algorithm

The proposed automatic algorithm was divided in the following steps:Create the stabilization diagram using the SSI algorithm and classify each pole based on stabilization criteria;Clear the stabilization diagram using the Hard Validation Criteria (HVC);Group poles that represent the same mode using agglomerative hierarchical clustering;Remove from each cluster poles of repeated orders, so that only one pole of this order remains;Eliminate small clusters that probably represent clusters of spurious or mathematical poles;Perform an outlier detection based on the boxplot method;Describe the global modes by the clusters mean frequency, mean damping, and mean mode shape;Group poles with mode shapes of high correlation using agglomerative hierarchical clustering.

In the following, the choice of all above-mentioned steps is justified and clarified.

#### 2.3.1. Stabilization Diagram and Stabilization Criteria

For the first step, it is possible to employ either SSI-COV or SSI-DATA algorithms, although the authors decided for the last. The identification is performed with increasing model orders and all extracted poles are inspected and classified according to the following evaluation. A k-order pole is stable if there is at least one (k−1)-order pole that satisfies the following stabilization criteria:(10)$$\Delta f_{m,n}=\frac{\left|f_{n}-f_{m}\right|}{f_{n}}<lim_{f},$$
(11)$$\Delta \zeta_{m,n}=\frac{\left|\zeta_{n}-\zeta_{m}\right|}{\zeta_{n}}<lim_{\zeta },$$
(12)$$MAC_{m,n}=\frac{{\left|\varphi_{m}^{H}\varphi_{n}\right|}^{2}}{\left(\varphi_{m}^{H}\varphi_{m}\right)\left(\varphi_{n}^{H}\varphi_{n}\right)}>lim_{MAC},$$
where m corresponds to the pole of order k under evaluation and n corresponds to any pole of order (k − 1). All limits are manually selected, but suitable values can be easily determined through initial analyses of the system. All poles that do not fulfill the stabilization criteria are classified as not stable.

#### 2.3.2. Hard Validation Criteria (HVC)

The idea behind the HVC is to remove all certainly spurious poles from the analysis of the following steps. In order to detect these poles, two criteria are employed: the damping ratio, and information about complex conjugated pairs.

As physical modes are characterized by positive damping ratios, it is expected that all poles with negative damping are spurious. Moreover, performing initial tests allow the analyst to know the normal behavior of the system, including the normal range of damping ratios. Thus, modes from the rotor or from its foundation are usually known within a determined range of damping. Furthermore, as already mentioned, rotating machines are constantly excited by periodic signals coming from their own operation or from the operation of other rotating parts in their surroundings. These harmonic frequencies appear in the stabilization diagram as stable poles of low or negative damping ratio due to their statistical aspects. Therefore, a first filter based on the poles damping ratio can be stablished as an HVC, all poles with values that are negative or out of the expected range being spurious, as employed by [6,8].

As mentioned by [6,7], every physical mode of a system appears in complex conjugated pairs, which makes it possible to classify as spurious all poles from the diagram that does not have a complex conjugated pair and remove them for the subsequent analysis.

#### 2.3.3. Agglomerative Hierarchical Clustering

Before applying the hierarchical clustering, some of the papers mentioned in the introduction added a step that would separate certainly spurious poles from probably physical ones using some criteria based on the pole’s mode shape complexity, such as Mode Phase Collinearity (MPC) and Mode Phase Deviation (MPD). As will be presented in the results, during the development of the algorithm proposed in this paper, it was observed that these criteria are not suitable to distinguish the rotor’s modes in the stabilization diagram. In order to avoid criteria that are not suitable for rotating machinery, it was decided to not apply a step before the hierarchical clustering.

As with all papers presented in the previous section, the machine learning technique called agglomerative hierarchical clustering was selected to group poles that represent the same mode. According to [7], the average linkage showed better results to create compact clusters of individual physical modes, thus it is used as the algorithm for hierarchical clustering. Moreover, in this paper, it was decided to employ the same algorithm with a similarity measure based solely on the frequency difference between all pair of poles, and the analysis of the MAC value within each cluster is postponed at the end of the algorithm.
(13)$$\Delta f_{m,n}=\left|\frac{f_{n}-f_{m}}{\max\left(f_{n},f_{m}\right)}\right|.$$

Since only the frequency difference is used as similarity difference, the threshold can be easily selected. This can be done from an analysis of the modes of interest variation in the stabilization diagram.

#### 2.3.4. Removal of Poles from Repeated Orders

In the case of closely spaced modes or spurious and mathematical poles near physical poles, it can be that more than one pole from the same order are grouped in the same cluster, which is not appropriate since each cluster is supposed to represent a single physical mode. Aiming to remove the repeated poles, a comparison of the damping ratio of each repeated pole with the cluster’s damping ratio median is done, given that no outlier removal was yet performed, and only the repeated pole with damping ratio closest from the median is maintained.

#### 2.3.5. Small Clusters Removal

Since physical modes tend to have a better stabilization when compared to spurious and mathematical poles (i.e., appear at several model orders in the stabilization diagram), the number of poles in each cluster can be used to separate clusters of spurious or mathematical poles from clusters of physical poles. A study by [7] presents a methodology that eliminates all clusters with sizes lower than 50% of the biggest cluster size. However, stabilization diagrams of rotating machinery data comprise both structural and rotor modes, the last being usually harder to stabilize in comparison with the first. Therefore, a 50% limit proved to exclude some rotors’ modes of interest from the analysis and the mean size of all clusters was adopted as a threshold.

#### 2.3.6. Outlier Detection

As a result of adopting just the frequency as a measure of similarity, a possible effect is that poles with different damping factors are grouped together in one cluster. As will be presented in the results, the SSI method is usually able to identify one of the closely spaced modes of the rotor (backward or forward) with an acceptable range of damping, whereas the other mode is identified with a lower or higher (or simply different) damping ratio. Given that these modes are closely spaced, it is possible that they end up grouped in the same cluster. In order to eliminate the modes with lower or higher (or simply different) damping, the outlier detection proposed by [5] is adopted. This approach was chosen because of the lack of information about the probability distribution of the clusters, and the outlier detection based on the quartile’s information results in a more conservative and effective method to remove outliers.

Furthermore, it is possible that a mode is identified in the stabilization diagram with high dispersion or with poles that, due to the order, end up far from the average of the mode. Aiming to maintain clusters with low frequency dispersion, the same approach adopted to detect damping ratio outliers is considered to detect frequency outliers. Thus, the outlier analysis is performed for both damping and frequency values.

#### 2.3.7. Global Modes

Finally, each cluster mean frequency, mean damping ratio, and mean mode shape are extracted to describe the global modes. The mean was adopted because an outlier analysis was performed, but it is also possible to use the pole with median damping ratio, as done by [6].

#### 2.3.8. Agglomerative Hierarchical Clustering of Each Cluster

Since the MAC value was not employed in the similarity measure of the third step and the mode shapes of each cluster were not evaluated in any other step of the algorithm, it is possible that poles with inaccurate mode shapes were included in the results, which would render the mean mode shape estimate also inaccurate. In order to remove these poles from the clusters, the hierarchical clustering algorithm can be once again employed as an additional step of the algorithm to improve the estimates, as described above.

In order to implement this step, the MAC value is computed between all poles within each cluster, resulting in one MAC matrix for each global mode extracted by the algorithm. The minimum value of each matrix is then identified and compared with the MAC limit of the stabilization diagram, informed by the user in the first step of this algorithm. If the minimum value of a cluster’s MAC matrix is above the limit ($MAC_{min}>lim_{MAC}$), it means that all mode shapes within this cluster have high correlation and, therefore, that the mean mode shape computed in the last step is adequate to represent the mode shape of that global mode. However, if the minimum value of a cluster’s MAC matrix is below the limit ($MAC_{min}<lim_{MAC}$), it means that not all mode shapes of this cluster have high correlation and that the mean mode shape is not adequate to represent the cluster. In this last case, another processing step is required to remove the poles with low correlation and obtain another set of poles with mode shapes that have high correlation between them and that can represent the mode shape of that global mode. In order to do that, hierarchical clustering is employed with a similarity measure equal to the inverse of the MAC between two poles of the global mode under analysis:(14)$$\Delta MAC_{m,n}=\frac{1}{MAC_{m,n}}.$$

This way, the two poles that have low correlation (low MAC value) will be distant from each other, whereas the two poles that have high correlation (high MAC value) will be close to each other. For the threshold value, the inverse of the MAC limit of the stabilization diagram is employed, so that the resulting clusters will comprise only the poles with MAC values above the limit. Then, the biggest cluster is identified and only the poles from this cluster are selected to represent the global mode.

Once this procedure is performed for all clusters from the previous step, the means of the frequencies, damping ratios, and mode shapes are once more computed to represent each global mode.

The resulting algorithm is summarized in Algorithm 1.
**Algorithm 1:** Proposed Algorithm.Inputs: Stabilization diagram (frequency, damping ratio, and mode shape), damping ratio limits (ζmin and ζmax), stabilization criteria (limf, limζ and limMAC), and similarity measure threshold (limD).Output: Global modes
Classify as stable all poles that satisfy the stabilization criteria and as not stable all remaining polesClassify as spurious all poles with damping ratio lower than ζmin or higher than ζmax (Hard Validation Criteria—HVC) or that do not appear with a complex conjugated pairExtract the number of stable poles (nme)Create a matrix of zeros D∈ℝnme×nmeFor m in [1,nme]:5.1.For n in [1,nme]: Compute the distance between the poles m and n (dm,n) using the relative distance between the natural frequencies of both poles and assign the result to the matrix D in the position (m,n)Apply agglomerative hierarchical clustering taking the distance matrix D as the method’s similarity measure and consider the informed threshold (limD)Extract the number of clusters obtained (nc)For c in [1,nc]:8.1.If cluster c has more than one pole of each order, remove all poles of each order but one, and keep the one with the damping ratio closest to the cluster’s damping ratio median8.2.Store the number of poles and each modal parameter (natural frequency, damping ratio, mode shapes and order) of the cluster cCreate a histogram of the number of poles in each clusterExtract the mean size of the clustersSelect the clusters whose size is bigger than the mean sizeCreate a boxplot of the frequency and of the damping ratioRemove the outliers: -If ωn<Q1freq−1.5 IQRfreq or ωn>Q3freq+1.5 IQRfreq, remove the pole n because it is a frequency outlier-If ζn<Q1ζ−1.5 IQRζ or ωζ>Q3ζ+1.5 IQRζ, remove the pole n because it is a damping ratio outlierBeing Q1 is the first quartile, Q3 the third quartile, and IQR the difference between the upper and lower quartilesExtract the parameters that represent the clusters: mean frequency, mean damping ratio, and mean mode shapeExtract the number of global modes (ngm)For i in [1,ngm]: 16.1.Extract the number of poles (np)16.2.Create a matrix of zeros DMAC−i∈ℝnp×np16.3.For m in [1,np]:16.3.1.For n in [1,np]:Compute the MAC value between the poles m and n and assign the result to the matrix DMAC−i in the position (m,n)16.4.Extract the minimum value of the matrix DMAC−i (mini)16.5.If mini<limMAC:16.6.Create a matrix of zeros Di∈ℝnp×np16.7.For m in [1,np]:16.8.For n in [1,np]:16.9.Compute the distance between the poles m and n according to Equation (14) and assign the result to the matrix Di in the position (m,n)16.10.Apply agglomerative hierarchical clustering taking the distance matrix Di as the method’s similarity measure and considering the informed MAC limit (1/limMAC)16.11.Select the poles from the biggest cluster to represent the global mode i16.12.Extract the parameters that represent the modal globe: mean frequency, mean damping ratio, and mean mode shape

### 2.4. Description of Datasets

#### 2.4.1. Test Rig with Hydrodynamic Bearings

The first data set used in this work was taken from a test rig with a rotor supported by hydrodynamic bearings, displayed on Figure 2. The system is basically composed of a rotating steel shaft (15 mm in diameter and 719 mm in length) supported by two hydrodynamic bearings (31 mm diameter, 18 mm length, 90 µm of radial clearance, and ISO VG32 oil at ambient temperature as working fluid) connected to an electric motor through a flexible coupling. In addition, the system has a hard disk and an electromagnetic actuator (used to insert different types of noise into the rotor). The experiments were carried out with the rotor operating with an angular shaft velocity of 75 Hz and four accelerometers installed in both bearings (two accelerometers for each bearing) were used to collect the vibration on the Y and Z directions.

During operation, rotating machines can be subjected to different types of excitation conditions that can facilitate or hinder OMA’s application. In order to investigate it, Ref. [25] performed the identification of a rotating system through OMA techniques and revealed that different test conditions influence the extracted parameters, ranging from non-identification to precise identification of modal parameters, which characterizes challenges to the automatic algorithm proposed here. Hence, it was decided to use more than one test condition. For that, the data was collected with different inputs and sampling frequencies, and during different periods of time, resulting in the tests displayed in Table 1.

An EMA analysis was also carried out though the Stepped Sine method to determine the modal parameters of the rotor supported by hydrodynamic bearings, so that their correct values were known for further validation of the proposed OMA algorithm. For this test, the rotor’s speed was 75 Hz. Two sets of tests were carried out with a step of 0.25 Hz, the first one with frequency range between 48 Hz and 58 Hz, in order to identify the first rotor’s mode, and the second one with frequency range between 200 Hz and 220 Hz, in order to identify the second rotor’s mode. To each test, 5 measurements were collected to compute mean values and diminish random errors. The results are displayed in Table 2. It is important to emphasize that the Stepped Sine method was able to identify two pairs of natural frequencies, each one containing the forward and the backward frequencies of the rotor, whose occurrence is traced back to the gyroscopic effect.

During the experiments, it was found that modal information of the foundation was transferred to the rotor’s dynamic response. A further modal analysis of the foundation was required so that the modal parameters extracted through OMA could be properly assigned to the system component that originated them. For the extraction of the foundation’s modal parameters, EMA was applied to the foundation after the shaft removal and with the use of FRF estimators and an impact hammer. The structure’s excitation was performed by means of impulses applied to the bearing housings in the Y and Z directions and the responses were measured using accelerometers mounted in the three directions (X, Y and Z) of the bearing housings. Frequency Response Functions (FRFs) were estimated, gathered, and evaluated only in the frequency range of interest (80 Hz to 320 Hz). The Least Square Complex Exponential (LSCE) algorithm was employed to estimate the modal parameters and the results are depicted in Table 3.

It is important to mention that although several foundation modes were identified, not all of them are excited during the rotor’s operation, which causes only a few to appear when applying modal analysis through the rotor’s vibration signals.

#### 2.4.2. Test Rig with Rolling Bearings

The second data set employed in this work was taken from the same test rig presented in Figure 2, replacing the hydrodynamic bearings by rolling bearings (15 mm inner diameter NJ 202 by NSK^®^) and using different excitation conditions. There are only minor variations in the positioning of each component due to the inherent inaccuracy of the assembly, disassembly, and alignment process of the system. The goal of these tests was also to evaluate the proposed algorithm in a system with lower damping, as expected for rolling bearings when compared to hydrodynamic bearings. Four accelerometers installed in both bearings (two accelerometers for each bearing) were again employed to collect the vibration on the Y and Z directions. The experiments were carried out with the rotor rotating in 30 Hz and under different operating conditions, resulting in the tests displayed in Table 4.

In order to evaluate OMA’s results, an EMA analysis was also carried out through the Stepped Sine method. For this test, the rotor’s speed was 30 Hz, and the test was carried out with frequency range between 20 Hz and 75 Hz and a step of 0.25 Hz, with the aim of evaluating only the first vibrating mode of the system. Two tests were carried out, one where the excitation was applied in the Y direction and other where the excitation was applied in the Z direction. The results are displayed in Table 5, where the values correspond to the obtained averages. 

As before, the Stepped Sine method was able to identify a pair of natural frequencies, containing the forward and the backward frequencies of the rotor. The significant reduction in damping values is noted when compared to the system supported by hydrodynamic bearings (compare Table 2 and Table 5). Regarding the small variations in the natural frequencies, these are more related to the inherent difficulty of positioning the components, as previously mentioned.

## 3. Results

The proposed algorithm is applied to two different datasets: response measurements of a test rig with a rotor supported by hydrodynamic bearings, and response measurements of a test rig with a rotor supported by rolling bearings.

The results obtained through the test rig with the rotor supported by hydrodynamic bearings are the first ones to be presented. To illustrate all steps of the algorithm, clarifying the analyzes performed by them, test 1 of Table 1 is taken as the standard example and a comprehensive explanation of its results is presented. Then, the algorithm is applied to all other tests in Table 1 and the main results are presented and discussed in order to show the algorithm’s robustness when different operating conditions are present.

Later, the test rig supported by rolling bearings, which has a higher stiffness and a lower damping when compared to the first test rig, is analyzed to verify the algorithm’s robustness to distinct systems. The results of all tests of Table 4 are briefly presented and discussed.

The algorithm, as well as the SSI-DATA method, were implemented in the programming language Python™.

### 3.1. Test Rig with Hydrodynamic Bearings

The stabilization limits considered in the following analysis were 0.2% for the frequency variation, 2% for the damping ratio variation, and 95% for the minimum MAC value, all of them conservatively chosen. The range [0.3%,10%] was used as the damping ratio limit. All stabilization diagrams were built with a maximum order of 100, with fixed 100 block rows.

Figure 3 displays the stabilization diagram of the first test of Table 1, excitation with white noise (medium intensity) and a sampling frequency of 2048 Hz. The diagram is presented in the frequency range of 0 Hz to 256 Hz, the range of interest in this analysis. From the diagram, one can observe that there are three alignments of spurious poles, the first at 75 Hz (the rotor’s rotating speed), two at 150 Hz (first harmonic), and the last at 225 Hz (second harmonic). The identification of the rotating speed and its harmonics as spurious was possible due to the HVC related to the damping ratio. In addition, several mathematical poles were also classified as spurious and, therefore, will not enter the following analysis. One can also observe that, close to the first rotor’s mode, two poles are predominantly identified in each order, which could lead to the idea that both forward and backward frequencies are identified. However, the second poles of each order are mostly identified with a high damping ratio (>7%), being inadequate to represent any rotor’s frequency.

The MPC (computed as described in [26]) and the MPD (computed as described in [6]) values of each pole were computed to perform additional analysis. The MPC value ranged from 63% to 99% for the first rotor’s mode, the highest ones (>86%) being outliers because of the high damping ratio (>5%), as will be seen in a further outlier analysis. For the second one, the range was 98% to 100%. For the foundation mode of 241.9 Hz, the values were much more stable, ranging from 94% to 98%. The MPD value, in contrast, ranged from 8% to 35% for the first rotor’s mode, the lowest ones (<19%) being outliers because of the high damping ratio (>5%). For the second one, the range was 3% to 6%. For the foundation mode of 241.9 Hz, the range was 11% to 16%. Therefore, if any clustering algorithm or HVC based on the MPC or MPD values were employed, the first rotor’s mode could be identified due to its great dispersion as spurious, and the identification algorithm would fail to provide reliable information.

After building the stabilization diagram and applying the HVC, the hierarchical clustering was performed. For the selected threshold definition, the distance between the known difference of closely spaced modes was employed. The difference between the first and second frequencies of the first mode, according to Equation (13), is 0.006. For the second mode, the difference is 0.002. Tests considering thresholds near these values were evaluated, resulting in a selected threshold of 0.01. It is important to emphasize that this threshold proved itself adequate for all other tests of Table 1, demonstrating how simple it is to select a value that works in different operating conditions of the same system. Figure 4 displays the obtained dendrogram, in which each cluster is represented by a different color in the bottom of the dendrogram and whose x-axis is organized with the frequency range of 53 Hz to 250 Hz, distributed in an ascending order.

Figure 5 displays the diagram of each cluster’s size, along with the limits proposed by this paper and by [7] to remove small clusters. From Figure 5, one can see that if the limit proposed by [7] was considered, the 6th and the 8th foundation modes would not be identified by the algorithm. There are also cases in which the first rotor’s mode is below the limit proposed by the authors, as the signals obtained from test 3 show. Therefore, the limit defined by the mean size is justified.

The outlier analysis was performed within the 10 clusters that remained from the previous analysis. Figure 6 displays the boxplot of both frequency and damping ratio values. Points out of the box range are considered outliers. Taking the first cluster as an example, which represents the first rotor mode, there are outliers in both frequency and damping ratio, although the first ones (53.16 Hz, 53.24 Hz, and 53.94) are less pronounced than the last ones (all damping ratios above 4%). From Figure 6, it is possible to see that the outlier analysis was adequate to remove outliers from all modes.

Concluding all essential steps proposed by the algorithm, the averages of the frequencies and of the damping ratios of the poles inside each cluster are extracted. The results are displayed in Table 6, along with the standard deviation of these parameters, the difference between the maximum and minimum values within the cluster that originated them, the errors in relation to the EMA references, the size of the cluster, and the lowest value in the MAC matrix, which will be further employed in the optional step to obtain sets of poles with high correlation mode shapes. From Table 6, one can see that most of the identified modes presented low standard deviations and low differences between maximum and minimum, for both frequency and damping ratio, and bigger cluster sizes.

It is important to mention that, although the first two modes of the rotor are composed by two frequencies, the backward and the forward ones (Table 2), the algorithm was not able to identify both of them. Since the similarity measure encompasses only the frequency difference between the poles, as presented in Equation (13), and considering the fact that the frequency and the damping ratio of the backward and forward frequencies are significantly close, it would be possible that both frequencies were grouped in the same cluster. However, the minimum MAC value for this mode was 98%, indicating a high correlation between all mode shapes within the cluster. Since some difference is expected from the mode shapes of the forward and backward frequencies, it is more likely that only poles of one of these frequencies are present in the cluster, indicating that the proximity of these two frequencies lead the SSI method to identify only one of them.

It is also important to mention that not only the rotor’s modes were identified, but also several modes from the foundation. Comparing Table 6 with Table 3, one can see that the modes identified with the OMA algorithm do not have the exact same parameters as the modes identified by EMA (but are relatively close). However, one must also recall that the EMA test was performed without the shaft and this variation of the modal parameters was already expected. Comparing the foundation’s results with the rotor’s results, one can observe that the errors were similar, highlighting the algorithms’ ability to extract accurate modal parameters for both the rotor and the foundation.

Moreover, Table 7 displays the errors between the EMA values and estimated values of the rotor’s modal parameters using the proposed algorithm, in which all but one parameter presented a low error. The highest error was on the damping factor of the first mode, whose occurrence can be traced back to the SSI method’s ability to estimate this parameter. Table 6 and Table 7 demonstrate the proposed algorithm’s capability of extracting the modal parameters of a rotating machine.

With the clusters of each global mode and the lowest value in their MAC matrices, the additional step of the algorithm can be performed. The modes of 53.4 Hz, 114.5 Hz, 139.9 Hz, 202.06, 212.2 Hz, and 243.1 Hz presented good results, since the minimum values on their MAC matrix were greater than the MAC limit of the stabilization diagram (95%), an expected value from poles from the same mode. Therefore, no alteration will be performed in the clusters of these modes. However, the other modes (158.1 Hz, 180.8 Hz, 191.5 Hz and 219.5) presented values lower than the MAC limit of the stabilization diagram. Hence, hierarchical clustering based on the MAC values was performed, obtaining, for each mode, a new set of poles from which the mean, the standard deviation, and the difference between the maximum and minimum values of the modal parameters were computed. The results are displayed in Table 8, from which one can verify that the minimum MAC value of all modes is now at least 95%, indicating that the obtained clusters present mode shapes with high correlation and, therefore, the mean of the mode shapes of each cluster is adequate to represent these modes. It is also possible to verify that no significant alteration occurred on the mean values of the modal parameters. In addition, the standard deviation and the difference between the maximum and minimum of most of the clusters achieved lower values (values highlighted in green), whereas only two modes exhibited an increase in standard deviation (values highlighted in red).

After these analyzes, the proposed algorithm, ignoring the additional step, was applied to all tests of Table 1 and the results obtained for the rotor’s modes are displayed in Table 9. As Table 9 shows, the proposed algorithm was able to extract the rotor’s modes from all tests, these having a small standard deviation and with mean values close to the values selected via EMA. It is important to mention that the main reason for the high errors in the damping ratio estimations is the low magnitude of this parameter. Moreover, the estimation of damping ratios is a challenge even when well consolidated EMA techniques are used for the modal identification, and high errors are also obtained when the results of different EMA techniques are compared. In this context, the estimations displayed in Table 9 are very good.

As occurred in Test 1, the application of the proposed algorithm to the remaining tests of Table 1 also enabled the identification of several foundation modes. In order to summarize the results, Figure 7 displays all modes estimated through the proposed algorithm as black dots, all rotor modes as continuous lines, and all foundation modes estimated by EMA as dashed lines. The frequency is presented in the x-axis and the data used to estimate the modes is presented in the y-axis. As indicated by Figure 7, most foundation modes were identified. Recalling the stabilization diagram of Figure 3, obtained with the data with medium intensity white noise, one can see that there are some frequency ranges in which the stabilization is irregular. Therefore, the absence of some foundation modes can be, once more, associated with the challenges in the SSI method.

With these analyses, an investigation was performed to evaluate the differences between dividing the hierarchical clustering in two steps, one based only on the frequency difference between the poles, and other based only on the MAC value, as proposed in this paper, and applying the hierarchical clustering in one single step, considering both the frequency difference and the MAC value, as proposed by other papers in the literature. In this case, the third step of the algorithm was modified so that the similarity measure comprised the frequency difference and the MAC value. Then, it was applied to all tests of Table 1, without the additional step, and considering four different threshold values (0.04, 0.06, 0.08 and 0.1). The results are displayed on Figure 8, along with the results from the proposed algorithm with the additional step to facilitate the comparison. In some cases, the modified algorithm identified global modes with very close frequencies. Due to the frequency range of Figure 8, these cases would not be visible. Therefore, the icons representing them have been modified, and are represented with solid icons rather than hollow ones.

From Figure 8, one can see that most of the frequencies identified by the proposed algorithm were also identified by the modified one. However, there are several cases in which two very close frequencies are identified, especially when the threshold of 0.04 is used. Analyzing the frequency range of the first rotor;s mode, one can see that the threshold of 0.04 identified two frequencies of approximately 53 Hz for Tests 1, 4, 5, and 6, and the thresholds of 0.06, 0.08, and 0.1 performed the same for Test 6. When the stabilization diagram of Figure 3 was analyzed, it was verified that this frequency range indeed presented the stabilization of two different modes. However, the damping factor of one of them made it inadequate to represent any rotor’s frequency. That is also the case for all other tests. Therefore, the identification of two frequencies near the rotor modes by the modified algorithm represents a disadvantage of using one single hierarchical clustering with similarity distance that comprises both the frequency difference and the MAC value. 

Evaluating other frequency ranges, it is possible to identify the same phenomenon in some foundation modes (124.8 Hz, 138.6 Hz, 157.4 Hz, 196.0 Hz and 204.0 Hz), mostly in the results from the modified algorithm (only the foundation modes of 124.8 Hz and 138.6 Hz of test 5 for the proposed algorithm). Analyzing each stabilization diagram, it was observed that most pairs of close frequencies were identified because poles from a single physical mode happened to be divided into more than one cluster by the algorithms due to irregularities in the stabilization diagram. The exceptions were the frequencies near 124.8 Hz of Tests 4 and 5, since the stabilization diagrams of these tests really present the alignment of two modes. However, it is possible that one of the alignments is actually an alignment of spurious modes rather than a closely spaced mode of the foundation, as occurred for the first rotor’s frequency.

Furthermore, there are some cases in which a foundation mode was identified by one of the algorithms and not by the other. These cases occurred 16 times, for both algorithms and all thresholds, and occurred for the foundation modes of 124.8 Hz (Tests 1 and 6), 157.4 Hz (Tests 1 and 3), and 196.0 Hz (Tests 4, 5 and 6). In five of these cases, the employed algorithm was the modified one with a threshold of 0.08. The modified algorithm with thresholds of 0.06 and 0.10 were responsible for three cases each, and the modified algorithm with a threshold of 0.04 was responsible for two cases. The proposed algorithm, in turn, was responsible for three cases.

Moreover, when the modified algorithm is employed, there is no guarantee that the minimum MAC value between the poles of a global mode is above the limit established for the stabilization diagram. Considering the global modes identified in all tests, when the threshold of 0.04 is used, 7 of the 74 identified global modes presented MAC values below 95%, with the minimum being 91%. When the threshold of 0.06 is used, 21 of the 66 identified global modes present values below 95%, with a minimum of 88%. When the threshold of 0.08 is used, 30 of the 62 identified global modes present values below 95%, with a minimum of 80%. Finally, when the threshold of 0.10 is used, 32 of the 64 identified global modes present values below 95%, with a minimum of 80%.

Considering the results presented here and that only one threshold value was selected for all tests of the proposed algorithm, some findings must be summarized. When the modified algorithm with low thresholds is used, there is a tendency to increase the division of poles belonging to the same physical mode into more than one cluster, which represents a disadvantage to the modal identification. If the threshold increased, the tendency decreases, but even when the threshold of 0.10 was used, the number of times that the division happened was higher than when the proposed algorithm was used. In addition, the increase of the threshold value proved to increase the number of global modes with a minimum MAC value below the limit of the stabilization diagram, and decrease these minimum values, which could lead to inaccuracies in the mode shapes’ mean. As to the non-identification of some foundation modes, both algorithms performed in the same manner. However, considering that the objective of this paper is the correct identification of the rotor’s modes, the identification of a spurious global mode near the first rotor’s

Frequency, along with the other findings, demonstrated the superiority of the proposed algorithm’s performance.

### 3.2. Test Rig with Rolling Bearings

To verify the robustness of the proposed algorithm, a distinct system will be analyzed. All data presented in Table 4 will be verified and the results will be briefly presented here, with focus on the identification of the rotor’s modes.

For the construction of the stabilization diagrams, the same stabilization and damping ratio limits and stabilization diagram parameters were considered throughout the results showed in this section. Figure 9 displays the stabilization diagram of Test 1 as an example. When compared to the one of Figure 3, this stabilization diagram shows fewer well-defined alignments of stable poles and more poles classified as not stable. However, it is also possible to identify in Figure 9 two well-defined alignments of stable poles near the rotor’s modes (Table 5), which, unlike the stabilization diagram of Figure 3, have modal parameters that make them adequate to represent both backward and forward frequencies. These particularities characterize this data set as a source of information about the modal parameters of closely spaced modes and as a real challenge to the identification of the foundation’s modes.

After building all stabilization diagrams, the algorithm follows by considering the threshold of 0.01 for the hierarchical clustering of all data sets, and the same one is used in the analyses from the previous section, demonstrating again how easy it is to select this threshold. The additional step was also considered to generate the results of the test rig supported by rolling bearings. The results for the rotor’s modes are displayed in Table 10, from which one can see that, even with unfavorable excitation conditions, the algorithm can extract representative global modes for the rotor, with low standard deviations and modal parameters close to the ones estimated by EMA.

Comparing Table 9 and Table 10, one can observe that the estimation’s errors are really close to each other, demonstrating the algorithms’ robustness to different datasets. 

As mentioned in the previous section, it is expected that the forward and backward frequencies present different mode shapes. Since the test rig supported by rolling bearings provided good results for both frequencies, their mode shapes were compared. Test 1 of Table 4 was once more taken as an example and the MAC value was computed between the mode shapes of all poles from the backward frequency and the mode shapes of all poles from the forward frequencies, producing a MAC matrix of 75 × 68 (the number of poles from the clusters of the backward and forward frequencies, respectively). The mean, maximum, and minimum MAC values of the matrix were 75%, 82%, and 67%, confirming the expected difference.

Moreover, in order to evaluate the ability of the proposed algorithm to extract the foundation’s modes when a different system is considered, Figure 10 displays the extracted modes as black dots, the rotor’s modes as continuous lines, and the foundation’s modes as dashed lines. From Figure 10, one can see that the algorithm was able to extract several of the foundation’s modes from the data of Test 1. When data from different tests are employed, only a few foundation’s modes are identified, which could be associated to unfavorable test conditions, and some modes out of the investigated frequency range (80 Hz to 320 Hz) appear. Moreover, the algorithm identifies some extra modes near the foundation mode of 157.4 Hz when data from Tests 1 and 2 are employed. Investigations performed with the same test rig by [25] detected a mode associated to the bearings housing near the frequency of 155 Hz, which would explain these extra identified modes. Therefore, the proposed algorithm demonstrated a good ability to identify the foundation’s modes.

As already mentioned, the results of this section were generated considering the additional step; however, the algorithm considering only the essential steps would also be capable of identifying accurate frequencies and damping ratios of all rotors’ modes, which was also observed in the results from the test rig supported by hydrodynamic bearings. Hence, the additional step is recommended when a higher precision in the mode shapes estimation is required or when a MAC criterion inside each cluster needs to be respected.

## 4. Conclusions

In this paper, a new automated algorithm to carry out modal parameter identification on rotating machinery through OMA is proposed. The novelty of the work is precisely the fact that it was developed for the identification of the rotor’s modes, and tested for unideal operating conditions that are usually present in the operation of rotating machines. The algorithm was applied through two datasets: vibration signals from a test rig with a rotor supported by hydrodynamic bearings and vibration signals from a test rig with a rotor supported by rolling bearings. Each step of the algorithm was presented, explained, and illustrated, highlighting the differences to other algorithms proposed in the literature, which were mainly developed to deal with signals from structures rather than from rotating machines. 

The test in which the operating rotor supported by hydrodynamic bearings is excited by the white gaussian noise of medium intensity was used to illustrate each step of the algorithm. From the results, it was possible to verify that some of the measures proposed by other papers to differentiate physical poles from mathematical and spurious poles are inadequate when the system under analysis is a rotating machine. The results of this data set and of the data sets with other excitation conditions also demonstrated that the proposed algorithm can extract from the stabilization diagram representative and accurate frequencies and damping ratios for both the rotor’s and the foundation’s modes, even when unfavorable test conditions are present.

Moreover, investigations were carried out to evaluate the performance of the algorithm when the additional step is implemented to the group, with hierarchical clustering and poles with high MAC values within each global mode. From the test with white gaussian noise of medium intensity excitation, the results showed that the additional step can find sets of poles with mode shapes of high correlation. The additional step was also compared with an algorithm that considers a single hierarchical clustering with similarity measure comprising both the frequency difference and the MAC value, as proposed by some previous authors. The results showed that the algorithm proposed in this paper, considering the additional step, presented better results than previous algorithms, especially when the correct identification of the rotor’s modes is considered.

When applied to a different system (a rotor supported by rolling bearing), the algorithm was also able to extract from the stabilization diagram representative and accurate frequencies and damping ratios for both the rotor’s and the foundation’s modes. These results demonstrated that the proposed algorithm maintained its robustness even when a different system was employed. In addition, the backward and forward frequencies of the first rotor’s mode were identified and the mode shapes extracted for each one confirmed that some difference between them is expected.

Therefore, the proposed algorithm proved to be an adequate and promising tool to extract modal parameters of rotating machines in operation. Further investigations are required to improve the extraction of representative mode shapes and the differentiation of the rotor’s backward and forward frequencies. 

The results were obtained by applying the proposed algorithm to data from test rigs. However, it is expected that it also works on more complex systems. The aim of the ongoing works is to test it in more complex systems, such as engines and compressors, to identify modes from both the rotor and the foundation. Once the algorithm’s robustness to more complex equipment is verified, the goal is to use it to monitor the modal parameters of the system and identify failures, given that variations in the modal parameters may be caused by them. With that, one can enable the SHM via OMA.

## Figures and Tables

**Figure 1 sensors-23-01665-f001:**
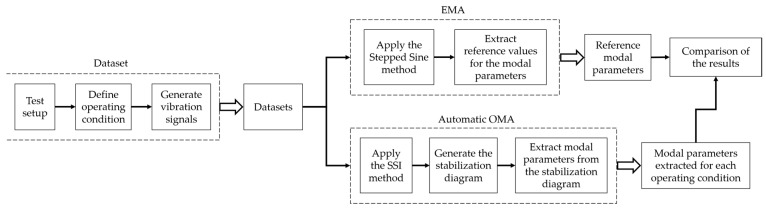
Diagram of the proposed approach.

**Figure 2 sensors-23-01665-f002:**
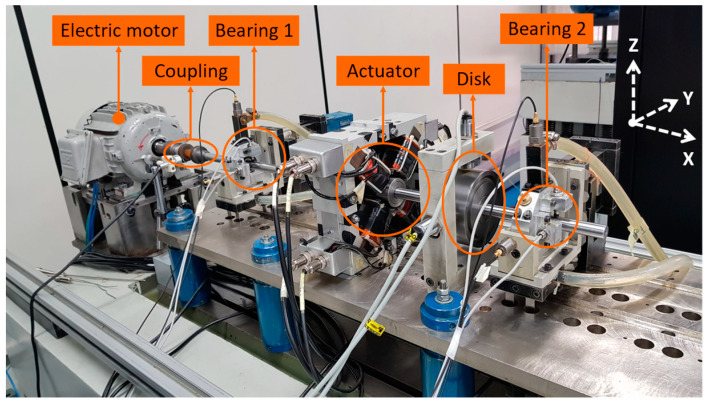
Test rig with hydrodynamic bearings.

**Figure 3 sensors-23-01665-f003:**
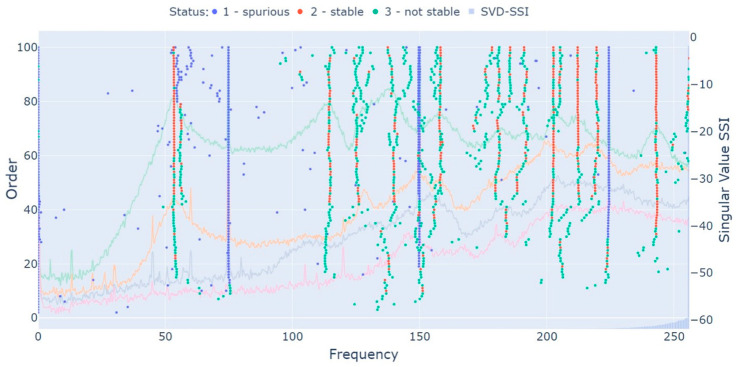
Test 1 (white noise—medium intensity) stabilization diagram.

**Figure 4 sensors-23-01665-f004:**
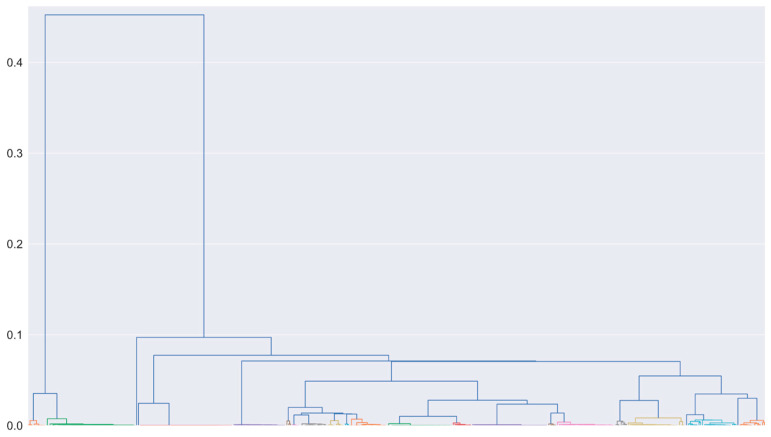
Test 1 (white noise—medium intensity) hierarchical clustering dendrogram.

**Figure 5 sensors-23-01665-f005:**
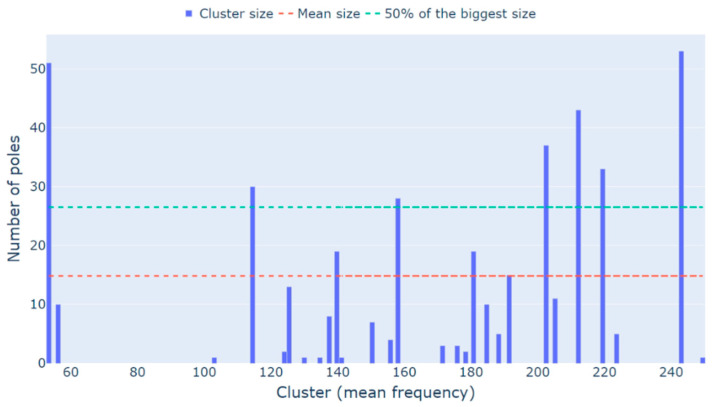
Test 1 (white noise—medium intensity) small clusters removal.

**Figure 6 sensors-23-01665-f006:**
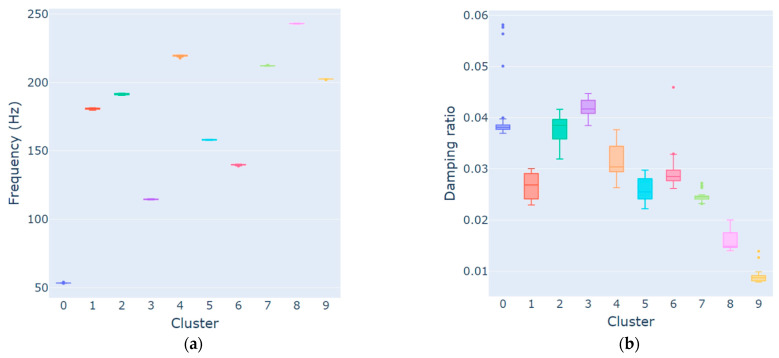
Test 1 (white noise—medium intensity) outlier analysis for frequency (**a**) and damping (**b**).

**Figure 7 sensors-23-01665-f007:**
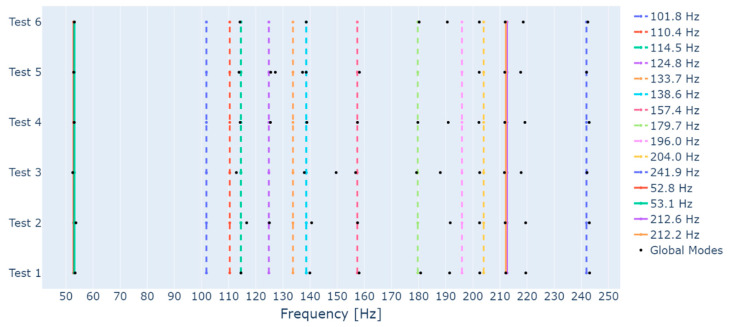
Foundation’s modes identified through the proposed algorithm.

**Figure 8 sensors-23-01665-f008:**
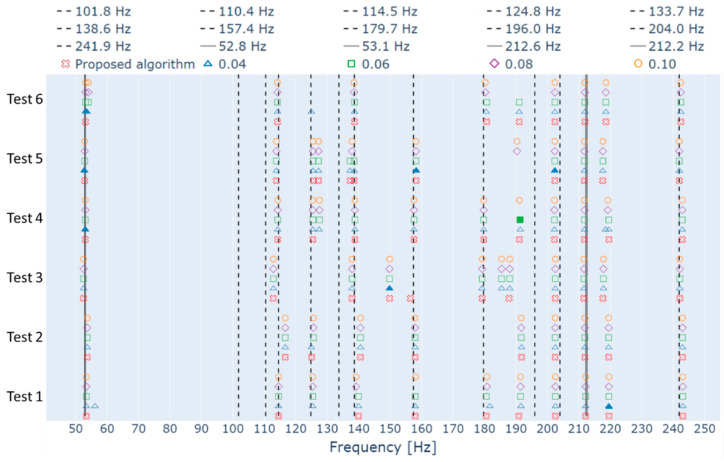
Foundation’s modes identified through the proposed and modified algorithms.

**Figure 9 sensors-23-01665-f009:**
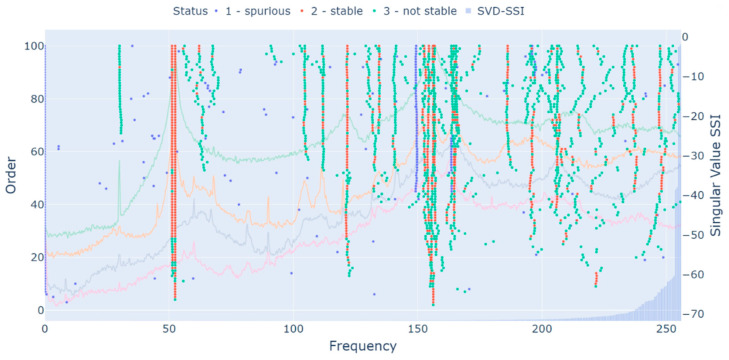
Test 1 (white noise—medium intensity) of the test rig with rolling bearings stabilization diagram.

**Figure 10 sensors-23-01665-f010:**
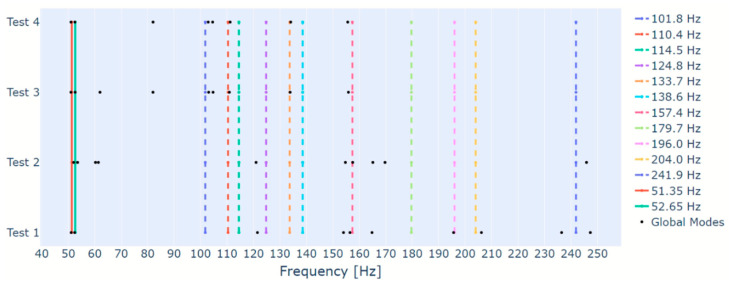
Foundation’s modes identified through the proposed algorithm for the test rig with rolling bearings.

**Table 1 sensors-23-01665-t001:** Test conditions for the test rig with hydrodynamic bearings.

Test	$f_{s}$ [Hz]	Time [s]	Excitation Direction	Excitation
1	2048	240	Y	White noise—medium intensity
2	2048	240	Y	White noise—low intensity
3	2048	240	Z	White noise and tapping
4	2048	240	Y	Blue noise
5	1024	240	Y	White noise—medium intensity
6	2048	120	Y	White noise—medium intensity

**Table 2 sensors-23-01665-t002:** Modal parameters of the rotor supported by hydrodynamic bearings.

Mode	Backward		Forward	
	Freq. [Hz]	Damp. [%]	Freq. [Hz]	Damp. [%]
First	52.8	4.26	53.1	4.25
Second	212.6	2.45	212.2	2.48

**Table 3 sensors-23-01665-t003:** Foundation’s modal parameters.

Mode	Freq. [Hz]	Damp. [%]
1	101.8	4.9
2	110.4	6.4
3	114.5	3.4
4	124.8	2.7
5	133.7	3.9
6	138.6	4.3
7	157.4	6.1
8	179.7	1.7
9	196.0	3.3
10	204.0	1.7
11	241.9	2.7
12	277.2	1.6
13	299.8	0.9

**Table 4 sensors-23-01665-t004:** Test conditions for the test rig with rolling bearings.

Test	$f_{s}$ [Hz]	Time [s]	Excitation Direction	Excitation
1	1024	60	Y	White noise—medium intensity
2	1024	60	Y	White noise—low intensity
3	1024	60	Y	White noise and tapping
4	1024	60	Y	Blue noise

**Table 5 sensors-23-01665-t005:** Modal parameters of the rotor supported by rolling bearings.

Mode	Backward		Forward	
	Freq. [Hz]	Damp. [%]	Freq. [Hz]	Damp. [%]
First	51.35	1.168	52.65	0.864

**Table 6 sensors-23-01665-t006:** Test 1 (white noise—medium intensity) test rig with hydrodynamic bearings global modes.

f [Hz]				ζ [%]				Size	MAC (Minimum)
Mean	Std.	$\Delta_{max,min}$	Error	Mean	Std.	$\Delta_{max,min}$	Error		
53.4	0.05	0.2	0.56%	3.81%	0.06%	0.28	11.55%	43	98%
114.5	0.26	0.9	0.00%	4.19%	0.18%	0.63	18.85%	30	99%
139.9	0.21	0.7	0.93%	2.84%	0.11%	0.37	51.41%	16	99%
158.1	0.18	0.7	0.44%	2.60%	0.23%	0.75	134.62%	28	73%
180.8	0.62	1.8	0.61%	2.65%	0.25%	0.71	35.85%	19	83%
191.5	0.53	1.5	2.35%	3.78%	0.30%	0.97	12.70%	15	91%
202.6	0.07	0.2	0.69%	0.87%	0.05%	0.20	95.40%	35	98%
212.2	0.10	0.4	0.00%	2.44%	0.04%	0.15	1.64%	37	99%
219.5	0.40	1.5	-	3.11%	0.26%	0.92	-	32	90%
243.1	0.16	0.7	0.49%	1.58%	0.18%	0.60	70.89%	53	97%

**Table 7 sensors-23-01665-t007:** Test 1 (white noise—medium intensity) rotor mode’s error.

Parameter	First Mode			Second Mode		
	EMA	OMA	Error	EMA	OMA	Error
f [Hz]	52.8	53.4	1.14%	212.6	212.2	0.19%
	53.1		0.56%	212.2		0.00%
ζ [%]	4.26	3.81	10.56%	2.45	2.44	0.41%
	4.25		10.35%	2.48		1.61%

**Table 8 sensors-23-01665-t008:** Test 1 (white noise—medium intensity) test rig with hydrodynamic bearings global modes after hierarchical clustering based on the MAC value.

f [Hz]			ζ [%]			MAC (Minimum)
Mean	Std.	$\Delta_{max,min}$	Mean	Std.	$\Delta_{max,min}$	
157.9	0.17	0.5	2.61%	0.13%	0.38	96%
180.6	0.81	1.8	2.62%	0.12%	0.41	95%
190.9	0.34	0.8	3.73%	0.18%	0.38	97%
219.6	0.52	1.5	3.28%	0.23%	0.64	95%

**Table 9 sensors-23-01665-t009:** Rotor’s global modes for the test rig with hydrodynamic bearings.

Mode		First Mode						Second Mode				
	f [Hz]			ζ [%]			f [Hz]			ζ [%]		
Test	Mean	Std.	Error	Mean	Std.	Error	Mean	Std.	Error	Mean	Std.	Error
EMA (backward)	52.8	-	-	4.26	-	-	212.6	-	-	2.45	-	-
EMA (forward)	53.1	-	-	4.25	-	-	212.2	-	-	2.48	-	-
1	53.4	0.05	0.56%	3.81	0.06	10.35%	212.2	0.10	0.00%	2.44	0.04	1.61%
2	53.7	0.09	1.13%	3.94	0.10	7.29%	211.9	0.05	0.14%	2.34	0.03	5.65%
3	52.5	0.09	1.13%	3.63	0.12	14.59%	211.6	0.15	0.28%	2.64	0.05	6.45%
4	53.1	0.05	0.00%	3.52	0.08	17.18%	211.8	0.19	0.19%	2.37	0.07	4.44%
5	52.9	0.12	0.38%	3.77	0.08	11.29%	211.8	0.11	0.19%	2.44	0.04	1.61%
6	53.1	0.02	0.00%	3.48	0.04	18.12%	211.9	0.15	0.14%	2.48	0.06	0.00%

**Table 10 sensors-23-01665-t010:** Rotor’s global modes for the test rig with rolling bearings.

Mode		Backward						Forward				
	f [Hz]			ζ [%]			f [Hz]	ζ [%]				
Test	Mean	Std.	Error	Mean	Std.	Error	Mean	Std.	Error	Mean	Std.	Error
EMA	51.35	-	-	1.168	-	-	52.65	-	-	0.864	-	-
1	51.12	0.03	0.45%	1.628	0.017	39.38%	52.43	0.00	0.42%	0.682	0.006	21.06%
2	52.07	0.12	1.40%	1.362	0.201	16.61%	53.55	0.02	1.71%	1.027	0.048	18.87%
3	51.04	0.04	0.60%	1.346	0.087	15.24%	52.62	0.01	0.06%	0.607	0.014	29.75%
4	51.06	0.03	0.56%	1.573	0.065	34.67%	52.51	0.04	0.27%	0.733	0.036	15.16%

## Data Availability

Not applicable.

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
