# Peer review of "Automated Operational Modal Analysis for Rotating Machinery Based on Clustering Techniques"

_sensors, 2023, doi:10.3390/s23031665_

Round 1

Reviewer 1 Report

This paper proposed an automated openerational modal analysis for rotating machinery based onn clustering techniques. A large number of experiments were carried out and the results were analyzed to prove the effectiveness and robustness of the method. However, there are some problems in the writing process:

An introduction to the literature should provide the author.

MAC should provide full name.

197 line formula number missing.

There are obvious format errors in Table 1.

Whether the “,” in the table should be“.”.

Line 567 has no table number.

Need to check the text of the figure number and table number is referenced correctly.

This paper need to be examined carefully. There are many times ' Error ! source Reference not found ' needs to be checked for correction.

Author Response

Attached are responses to all comments raised by reviewers.

Reviewer 2 Report

1.     As per the title of the paper, this paper is related to “Automated Operational Modal Analysis for Rotating Machinery”. But the introduction is emphasize structure health monitoring (SHM). Kindly rewrite and update the introduction with focusing “Automated Operational Modal Analysis for Rotating Machinery”.

2.     Need of study is missing. Kindly include one paragraph in the introduction

3.     Novelty of work is missing. Kindly include all novelties of the work done in this paper. List out all novelties in the bullet points in the second last paragraph of the abstract.

4.     Also rewrite the abstract after incorporation of novelties

5.     Also rewrite the conclusion after incorporation of novelties

6.     The organization of the paper is missing. Kindly include one paragraph n the end in the introduction.

7.     The proposed approach is missing. Kindly include a separate heading and explain in detail. Also include a diagram in this section that can represent the all steps of the whole work done in this paper.

8.     What is the reason behind to opt proposed method (Clustering Techniques) only in this study? Kindly justify the reason to select Clustering Techniques. Is this 1st time implementation in this research domain? Kindly validate with existing literature.

9.     The actual work is related to bearing diagnosis as shown in Figure 1. Kindly specify the detailed information of the testsetup and its different operating condition to record the data for this study. Also explain the bearing faulty and healthy condition and operating condition in different loading scenario

10.  The proposed work is based on mechanical signal analysis. But mechanical signal are prone to have high SNR (signal-to-noise ratio). In this situation what kind of methodology has been adopted to capture pure mechanical signals and remove the noise from the signal after capturing

11.  What kind of modal parameter identification is performed using mechanical signals. Kindly specify 1st and explain in detail in the section of the proposed approach

12.  As per the statement in the abstract two different datasets are used to validate the performance of the proposed approach. Kindly include a comparative table for results analysis and show the performance accuracy for both dataset

13.  The proposed approach is validated for different systems and test conditions. But different systems and test conditions are missing. Kindly include in a tabular form in the section of the proposed approach.

14.  As per the statement in the paper, the proposed algorithm is suitable for the accurate extraction of frequencies and damping ratios from the stabilization diagram, for both the rotor and the foundation. Kindly mention its correlation analysis or pattern analysis w.r.t the different conditions and operation of the rotating machine

15.  Kindly include results validation by implementing one more other method as well as comparing with existing literature.

Author Response

(The authors gave the same response as above.)

Reviewer 3 Report

The authors have proposed a new automated algorithm to carry out the modal parameter identification on rotating machinery through OMA. The topic of the paper and the presented results are interesting. The paper is very well prepared and is appropriate for the Sensors, MDPI journal. I think that the proposed algorithm has the potential for practical usage. I have some notes for the authors:
1. Please revise the mistake on page 9: 16.9. Compute the distance between the poles according to equation Error! Reference source not found.
Also on page 13: The difference between the first and second frequencies of the first mode, according to eq. Error! Reference source not
found., is 0.006...
2. A lot of variables are used in the paper. The list of abbreviations or/and nomenclature could be provided at the end of the paper.
3. It is not clear, why Python was used for algorithm implementation and whether the algorithm is developed for free usage (e.g. available on git-hub). What is the portability of proposed .py scripts?
4. What is the applicability of the designed diagnostic system to other types of engines and equipment? Please discuss it more.
5. The authors have used a decimal comma and decimal dots in tables (see Tab 3 and 4). Please check it. In English, only a dot is usually used as a separator.
6. In discussion, the practical use of the proposal should be more emphasized.
I suggest accepting the paper after minor revision.   

Author Response

(The authors gave the same response as above.)

Round 2

Reviewer 1 Report

All comments have already been incorporated. 

Reviewer 2 Report

-most of comments have already been incorporated. only one minor comment is required to update as:       

Title of the manuscript is required to update as of now it is showing that the proposed method is applicable for all rotating machines, but work is related to only some specific type of application.